# Dietary Polyphenols as Anti-Aging Agents: Targeting the Hallmarks of Aging

**DOI:** 10.3390/nu16193305

**Published:** 2024-09-29

**Authors:** Ying Liu, Minglv Fang, Xiaohui Tu, Xueying Mo, Lu Zhang, Binrui Yang, Feijie Wang, Young-Bum Kim, Cheng Huang, Liang Chen, Shengjie Fan

**Affiliations:** 1School of Pharmacy, Shanghai University of Traditional Chinese Medicine, Shanghai 201203, China; lydiaaaaa_liu@163.com (Y.L.); chuang@shutcm.edu.cn (C.H.); 2Nutrilite Health Institute, Amway (Shanghai) Innovation and Science Co., Ltd., Shanghai 201203, China; 3Division of Endocrinology, Diabetes, and Metabolism, Beth Israel Deaconess Medical Center, Harvard Medical School, Boston, MA 02115, USA

**Keywords:** anti-aging, polyphenol, the hallmarks of aging

## Abstract

**Background:** Aging is a natural biological process influenced by multiple factors and is a significant contributor to various chronic diseases. Slowing down the aging process and extending health span have been pursuits of the scientific field. **Methods:** Examination of the effects of dietary polyphenols on hallmarks of aging such as genomic instability, telomere attrition, epigenetic alterations, loss of proteostasis, disabled macroautophagy, deregulated nutrient-sensing, mitochondrial dysfunction, cellular senescence, stem cell exhaustion, altered intercellular communication, chronic inflammation, and dysbiosis. **Results:** Polyphenols, abundant in nature, exhibit numerous biological activities, including antioxidant effects, free radical scavenging, neuroprotection, and anti-aging properties. These compounds are generally safe and effective in potentially slowing aging and preventing age-related disorders. **Conclusions:** The review encourages the development of novel therapeutic strategies using dietary polyphenols to create holistic anti-aging therapies and nutritional supplements.

## 1. Introduction

Based on data from the World Health Organization (WHO Fact Sheet) and the World Bank (World Bank Data), the global population aged 65 and older was 727 million, constituting 9.3% of the total population in 2022. In China specifically, China’s population aged 65 and older accounted for 13.5% in 2020, an increase of 4.6 percentage points from 8.9% in 2010, and the elderly population is growing at a rate of 5% per year [1]. A country or region is deemed to enter an aging society when the proportion of the population aged 60 and older reaches 10% or that of the population aged 65 and older reaches 7%. Although it is controversial whether aging is an unavoidable degradative biological process, aging increases the risks of diseases and death, including metabolic disorders, cancer, and neurodegeneration [2]. Aging and age-related diseases significantly contribute to the health and financial burden worldwide as the increased elderly population.

Aging is complex, multifaceted, and influenced by genetic, environmental, and lifestyle factors [3]. In 2023, C López-Otín’s group expanded the original 9 hallmarks of aging to 12, providing a more comprehensive framework for anti-aging studies [4,5]. Currently, the most promising geroprotectors are targeting these hallmarks [6], such as genomic instability, telomere attrition, epigenetic alterations, loss of proteostasis, disabled macroautophagy, deregulated nutrient-sensing, mitochondrial dysfunction, cellular senescence, stem cell exhaustion, altered intercellular communication, chronic inflammation, and dysbiosis. For example, metformin is primarily known as an anti-diabetic drug. However, studies have shown that low-dose metformin can activate the AMPK pathway through the lysosomal pathway and exert a life-extending effect [7]. Nicotinamide mononucleotide (NMN), the precursor of NAD^+^, is evidenced by studies that show its role in maintaining high NAD^+^ content and exerting life-extending effects by activating Silent information regulator 1 (SIRT1), enhancing mitochondrial function and reducing oxidative stress [8]. Quercetin is a flavonoid mainly present in onions. The combination of quercetin and dasatinib has been proven to reduce intestinal senescence and inflammation in the elderly and improve health by ameliorating microbial dysbiosis [9]. Oral administration of spermidine, a natural polyamine, has been evidenced to delay brain aging by regulating the level of autophagy proteins and preventing apoptosis and inflammatory responses [10]. These findings indicate that targeting the hallmarks of aging is promising for alleviating and reversing aging.

Polyphenols, a primary group of phytochemicals found in plant-based foods, are garnering increasing attention for their anti-aging effects. Numerous studies have indicated that polyphenols possess a range of biological activities, including anti-inflammatory effects, promotion of cellular repair, and antioxidant capability [11,12]. Free radicals are unstable molecules that cause oxidative stress, leading to cellular damage and inflammation and ultimately contributing to aging. Polyphenols, acting as powerful scavengers, protect cells from oxidative stress, reducing the risk of age-related diseases such as heart diseases [13], cancer [14], and neurodegenerative diseases (NDs) [15]. Beyond their antioxidative prowess, polyphenols can also regulate immune function, inflammatory response, and oxidative stress and improve the resistance to diseases including systemic lupus erythematosus [16], rheumatoid arthritis [17], and multiple sclerosis [18]. In the aspect of the nervous system, polyphenols have potential protective effects on cognitive functions and alleviate NDs through enhancement of mitochondrial function, prevention of cellular damage, and proper functioning of the DNA repair pathway [19]. At the same time, polyphenols also play an important role in regulating metabolic processes, such as improving blood glucose and blood lipid levels [20]. For example, polyphenol-rich strawberry juice has hypoglycemic and hypolipidemic effects on streptozotocin-induced diabetic rats by restoring carbohydrate metabolism enzymes and antioxidant enzyme activities to the normal level and reducing lipid peroxidation and proinflammatory cytokines [21]. Overall, polyphenols from natural products have many benefits targeting the anti-aging process on various organs and tissues, such as the brain, muscle, skin, heart, liver, and intestine (Figure 1).

This review synthesizes the anti-aging properties of different classes of polyphenols, emphasizing the targeting on the hallmarks of aging, aiming to offer strategies for postponing the aging process and extending good health.

## 2. Biological Activities of Polyphenols

Polyphenols are a class of secondary metabolites widely found in plants with two or more phenolic hydroxyl groups. Polyphenols are particularly abundant in fruits, vegetables, tea, and red wine. Depending on their chemical structure, polyphenols can be classified into flavonoids, phenolic acids, stilbenes, lignans, and other polyphenols. Flavonoids possess functions such as antioxidation [22] and anti-inflammation [23]. Anthocyanins have the ability to be antioxidative and are beneficial to aspects such as vision [24]. Stilbenes have a certain anti-cancer effect [25]. Lignans have an impact on hormone regulation [26]. Among polyphenols, flavonoids are probably the most well-researched and well-known group. According to their composition, flavonoids can be further classified into subclasses such as flavones, flavanols (flavan-3-ols), isoflavones, flavanones, anthocyanins, and chalcone [27]. Flavonoids are valued for their strong antioxidant activity, which is promising to slow the aging process by scavenging free radicals and reactive oxygen species (ROS) and alleviating cellular damage caused by oxidative stress. It can also increase blood circulation, lower cholesterol, and lower cardiovascular disease risk [28,29]. In recent years, many different types of polyphenols have been found to have significant antioxidant properties and may play a role in preventing age-related chronic diseases.

As aging progresses, the risk of various metabolic disorders increases. Dietary polyphenols could be a novel and potential health element for regulating glucose and lipid metabolism [30]. Studies have shown that an additional intake of 500 mg of flavonoids per day can reduce the risk of type 2 diabetes [30]. Metabolic-associated fatty liver disease (MAFLD), for instance, is the most common chronic liver disease globally. Studies have shown that certain polyphenols, such as curcumin, which belongs to the curcuminoids and is found in turmeric, may play a role in mitigating the effects of aging on metabolism. Jalali et al. [31] discovered through meta-analysis that curcumin supplements can reduce the activities of ALT and AST, as well as TC, LDL, FBS, HOMA-IR, and waist circumference. Furthermore, it has been experimentally proven that curcumin can lower the plasma lipid level and modify the lipid metabolism of C57BL/6 male mice with high-fat, high-fructose-induced NAFLD and alter the activity of cytochrome CYP3A and CYP7A both in vitro and in healthy rat models [32]. Experiments have proved that resveratrol, a stilbene abundantly present in grapes, can increase the number of mitochondria and citrate synthase activity [33]. In insulin-resistant rodents, resveratrol promotes intracellular glucose transport and enhances the ability of skeletal muscle to absorb insulin-stimulated glucose [34,35]. (-)-Epicatechin is a natural flavanol that can be used for preventing cardiovascular risk diseases. Studies have shown that (-)-epicatechin can increase the expression of apelin/APLNR and improve lipid abnormalities and lipid catabolism disorders in offspring caused by maternal obesity [36]. This suggests that dietary polyphenols may have the potential to modulate glucose and lipid metabolism, which is often disrupted during the aging process and contributes to the development of age-related metabolic disorders.

Polyphenols can also modulate immune system function by inhibiting NF-κB and MAPK/ERK pathways, inhibiting toll-like receptors (TLRs) and pro-inflammatory gene expression, suppressing the enzymes related to the production of ROS, and up-regulating other endogenous antioxidant enzymes [37]. As a kind of flavanol, epigallocatechin-3-gallate (EGCG) could affect the differentiation of immune cell CD4^+^ T-cell subsets and alleviate experimental autoimmune encephalomyelitis (EAE) [38]. Research has demonstrated that resveratrol can target sirtuin, adenosine monophosphate kinase, NF-κB, inflammatory cytokines, and antioxidant enzymes to inhibit the expression of TLRs and pro-inflammatory genes, thereby regulating the immunity of the organism [39]. Apigenin, a dietary flavonoid mainly derived from celery and chamomile, can suppress the autoantigen presentation and stimulation functions requisite for the activation and expansion of autoreactive Th1 and Th17 cells as well as B cells in systemic lupus erythematosus (SLE), an autoimmune disorder, by down-regulating the expression of COX-2 [40].

Oxidative damage is one of the important reasons for cellular aging and functional decline. Resveratrol, which has a significant connection with anti-aging, can remarkably extend the lifespan of several model organisms by regulating oxidative stress, energy metabolism, nutrient sensing, and epigenetics [41]. Resveratrol has been shown to lower anti-apoptotic and antioxidant capacity-related genes to delay ovarian aging in the oxidative stress model [42]. Studies have shown that EGCG can decrease the acetylation level of histones by reducing the activity of histone deacetylase (HDAC) and further regulate the expression of genes related to cell aging and anti-oxidative stress, thereby delaying aging [43]. Catechin is a kind of flavanol with a relatively high content in green tea, wine, and cocoa products. It can not only scavenge free radicals and metal ion chelators but also participate in various enzymatic reactions that detoxify some peroxides in cells, thus preventing diseases caused by stress [44]. Cocoa flavanols act as an activator of Nrf2 to increase the level of antioxidant proteins, protect against skeletal muscle damage, increase the abundance of mitochondrial proteins, and improve the walking ability of patients with peripheral artery disease [45]. Although the metabolism levels of polyphenols in the brain are low, flavonoids have been demonstrated to have neuroprotective and cognitively enhancing benefits by lowering oxidative stress and neuroinflammation. Flavonoids may potentially improve the cerebrovascular system, regulate adult neurogenesis, restore healthy brain function, and slow brain aging [46].

The bioavailability of polyphenols in the gastrointestinal tract is relatively low, mainly accumulating in the large intestine and being metabolized by the gut microbiota that transform intestinal polyphenols [47]. The interaction between polyphenols and gut microbiota is mutual. Taking the lipophilic polyphenol curcumin as an example, curcumin is mainly detected in the intestinal tract after oral administration. On the one hand, human gut microbiota such as *Escherichia coli*, *Bifidobacterium longum*, *Bifidobacterium pseudocatenulaum*, and *Enterococcus faecalis* can convert curcumin into other metabolites through acetylation, hydroxylation, reduction, and demethylation [48]. The metabolites of curcumin, such as demethylcurcumin and bisdemethoxycurcumin can increase the activity of Aβ degrading enzymes and reduce the accumulation of Aβ in the hippocampus and cortex of AD model mice. This neuroprotective effect is not characteristic of curcumin itself [49]. On the other hand, the antibacterial activity of polyphenols enables them to increase the proportion of beneficial bacteria such as *Bifidobacteria*, *Lactobacilli*, and butyrate-producing bacteria in the gut microbiota and reduce the number of pathogenic bacteria such as *Prevotellaceae*, *Coriobacterales*, *Enterobacteria*, and *Rikenellaceae* [48]. The amelioration of gut microbiota balance contributes to upholding the integrity of the intestinal barrier, curtailing the influx of harmful substances into the organism, and abating the occurrence frequency and intensity of inflammatory responses. In vitro studies have shown that in Caco-2 cells, curcumin pretreatment significantly attenuates the inflammatory factor IL-1β secreted by intestinal epithelial cells and macrophages induced by cyclic lipopolysaccharide, reduces intestinal barrier dysfunction, and regulates chronic inflammatory diseases [50]. Urolithin A (UA) is the gut microbiota metabolite of phenolic acids ellagic acid (EA). Studies have shown that UA can activate the Nrf2 pathway to enhance the gut barrier integrity [51]. Oral administration of EGCG can significantly enrich short-chain fatty acid-producing bacteria and enhance anti-inflammatory effects and gut barrier integrity [52]. Consuming 1391 mg/day of polyphenols for 8 weeks can decrease the serum connexin level and strengthen the intestinal barrier function in elders over 60 [53].

Taken together, polyphenol is an important category of plant-derived bioactive substances that positively impact human health in multiple ways, such as regulating the body’s metabolism, enhancing immune function, and delaying aging (Figure 2). Having a reasonable diet and consuming more fruits and vegetables rich in polyphenols can help improve health conditions and quality of life.

## 3. Natural Polyphenols for Anti-Aging Studies

Polyphenols are acknowledged as an essential part of the human diet, and epidemiological data indicate that consuming a diet high in antioxidant fruits and vegetables lowers the incidence of numerous oxidative stress-related disorders, including diabetes, cardiovascular disease, and cancer. Polyphenols show great antioxidant and anti-inflammatory properties, are also easily absorbed in the gut, and constitute a significant supply of antioxidants in the diet [54]. Variations in the antioxidant capacity of particular phenolic compounds are caused by the quantity and location of their hydroxyl groups. Dietary polyphenolic components can extend the lifespan of various model species by removing senescent cells, maintaining mitochondrial homeostasis, suppressing inflammatory response, restricting calories, increasing autophagy, and countering oxidative stress [12,55,56,57] (Figure 2). It may control energy metabolism, extending lifespan and good health while lowering the chance of age-related chronic diseases. Increasing evidence has demonstrated that dietary polyphenol intake may delay aging and treat age-related diseases (Table 1).

### 3.1. Ellagic Acid (EA)

EA is widely found in fruits and nuts, especially pomegranates, raspberries, and walnuts, and its content is considerable. There is also a general understanding that eating pomegranate can fight against aging. Several studies have demonstrated that EA has strong biochemical and biological activity, such as antioxidative, anti-inflammatory, and neuroprotective properties, and its metabolites also play a role in anti-aging [116]. EA can increase the longevity of *Drosophila melanogaster*, *C. elegans*, and mice by scavenging free radicals, activating antioxidant enzymes, and decreasing the production of ROS [116]. Studies have shown that EA increases resistance to UV radiation, oxidative stress, and *Pseudomonas aeruginosa* infection stress in *C. elegans* to extend lifespan. It also activates stress-related genes through the insulin-like growth factor-1 (IGF-1) signaling pathway to reduce DNA damage and enhance resistance to UV radiation [75].

EA can be converted to urolithin-A (UA) in the gut via the gut microbiota. UA is a novel natural chemical that has been proven to stimulate mitophagy both in vitro and in animals following administration and has a favorable safety profile. Despite a great deal of research on the benefits of EA in extending lifespan, it has been proposed that the effects of EA are mediated through its metabolite UA. With aging, UA promotes mitochondrial autophagy, maintains mitochondrial biogenesis and respiratory capacity in *C. elegans*, and extends lifespan [82]. Clinical research has demonstrated that after 28 days of oral treatment of 500 or 1000 mg of UA, mitochondrial activity of the elderly is significantly improved [79]. Additionally, UA stimulates mitochondrial autophagy and enhances muscular health. In conclusion, further research is still needed to determine whether EA or its metabolites contribute to extended lifespan.

### 3.2. Gallic Acid (GA)

GA is widely present in various plants, such as tea, pomegranate, grapes, and nuts. It possesses a broad spectrum of antioxidant, anti-inflammatory, antibacterial, antiviral, and antitumor properties. GA could protect against mitochondrial oxidative stress and inhibit DNA damage and apoptosis. Studies have shown that GA protects mitochondria against extreme oxidative stress, increasing the survival of neuronal cells in a mouse model of hypoxia/reperfusion brain injury [117]. As the most abundant and biologically active tea polyphenol, EGCG treatment attenuates neuronal apoptosis and neurodegeneration, ameliorates oxidative damage, and protects against neurodegeneration in rat brain [118]. Additionally, UV radiation is a key cause of photoaging in the skin, resulting in collagen loss and wrinkle formation. GA could stimulate collagen synthesis to enhance skin elasticity while reducing wrinkle formation [119].

### 3.3. Rutin

The natural flavonoid rutin, also known as quercetin-3-O-rutinoside and vitamin P, possesses relatively stable physicochemical properties and is widely distributed in citrus fruits and buckwheat seeds [120]. Rutin has been demonstrated to extend the lifespan and healthy lifespan of *Drosophila melanogaster*, *C. elegans*, and mice due to its antioxidant and anti-inflammatory effects. It exhibits a remarkable ability to selectively target senescent cells and efficiently suppress the production of CXCL8, which is a typical SASP effector, in senescent cells. Compared with the SASP-specific targeted medications rapamycin and metformin, rutin has a higher security profile as a natural product [121]. Furthermore, diabetes is a chronic disease associated with oxidative stress. Supplemented with rutin on a regular basis, it enhances the quality of life for those with type 2 diabetes mellitus by lowering blood pressure and elevating antioxidant enzyme levels [122]. Environmental pollutants phthalates and bisphenol A are detrimental to cardiovascular health, and rutin supplementation upregulates Nrf2 and inhibits the NF-κB signaling pathway to prevent toxic-induced oxidative stress and inflammation in the heart [123].

### 3.4. Quercetin

Quercetin is widely found in a variety of fruits and vegetables, such as onions, apples, berries, and broccoli. Quercetin is a potent antioxidant with anti-aging properties that scavenge free radicals and shield cells from oxidative stress-related cell damage. It also lessens the harm that inflammation causes to tissue and suppresses inflammatory reactions. Furthermore, quercetin postpones illnesses linked to aging and safeguards the neurological and cardiovascular systems.

Senolytics are a class of drugs that can slow down the physical dysfunction caused by senescent cells and selectively eliminate senescent cells [124]. Dasatinib, which removes senescent human adipocyte progenitors, and quercetin, which kills senescent human endothelial cells and mouse bone marrow stem cells, work together to provide a significant anti-aging effect [125]. As the first combination of drugs in the Senolytics regimen, they are able to specifically target and potently eliminate senescent cells [97]. Senolytic treatment with dasatinib and quercetin decreases intestinal cellular senescence and inflammatory burden while modulating gut microbiome in aged mice, alleviating intestinal senescence and inflammation [9]. Senolytics are employed not only to lessen the burden that senescent cells produce but also to prevent the physiological dysfunctions they can induce.

Frailty is a geriatric syndrome induced by multiple age-related physiological changes [126]. A lower chance of frailty is linked to a higher intake of dietary flavonols, particularly in quercetin. Research indicates that a daily increase of 10 mg in quercetin intake lowers the risk of frailty by 35% [127]. Additionally, flavonols can benefit skin health. Senescent fibroblast accumulation, chronic inflammation, and collagen remodeling in the skin lead to age-related skin aging, characterized by wrinkles and a loss in skin elasticity.

### 3.5. Fisetin

Fisetin is a unique antioxidant mostly present in strawberries and other fruits and vegetables. It is one of the rarer flavonoid components that may cross the blood–brain barrier [128]. Fisetin has a chemical structure similar to quercetin, except for a hydroxyl group in position 5. However, compared to quercetin, fisetin has greater anti-aging properties [129]. Fisetin not only has direct antioxidant activity, but it also increases glutathione levels, the primary antioxidant in cells. In oxidative stress, fisetin preserves mitochondrial function and protects neuronal cells from oxidative stress-induced death. Fisetin has been demonstrated to improve skin aging by selectively eliminating senescent dermal fibroblasts and blocking SASP release, making it a promising novel anti-aging medication [130]. Fisetin has also been proven to ameliorate cognitive impairment in rats and extend long-term memory [131,132]. Fisetin has the potential to slow down brain aging and skin aging and may lower the risk of age-related neurodegenerative illnesses.

### 3.6. Anthocyanins

Anthocyanins are a type of natural pigment in a wide range of foods, particularly berries such as black, blue, and raspberries, with powerful antioxidative effects that help enhance blood vessel health, prevent cancer, and maintain strong bones [133]. Despite comparatively poor bioavailability, anthocyanins are more readily accessible for disease prevention due to their fewer adverse effects and lower cost [134]. Oxidative stress is a significant factor in the aging process. As an intracellular lysosome-dependent metabolic process, autophagy removes damaged cells and proteins to prevent inflammatory responses and maintains physiological homeostasis. Bilberry anthocyanin has strong anti-inflammatory and antioxidant effects, which have been demonstrated to induce autophagy via the AMPK–mTOR signaling pathway, further enhancing resistance to oxidative stress and slowing down the aging process [135]. Oxidative stress also leads to reduced skin collagen synthesis and elevated levels of several pro-inflammatory cytokines, which can lead to wrinkles and plaques. The necessity of maintaining good skin health is becoming more widely recognized, and anthocyanins are one of the natural antioxidants that can be readily ingested from food. Carboxypyranocyanidin-3-O-glucoside has potent anti-aging properties for the skin because it not only scavenges ROS but also stimulates the activity of endogenous antioxidant enzymes such as glutathione peroxidase, catalase, and superoxide dismutase [136]. Additionally, its excellent structural stability is advantageous when it comes to cosmetic preparations for long-term storage.

## 4. Polyphenols and Hallmarks of Aging

In recent years, polyphenolic compounds have attracted much attention due to their potential anti-aging effects. Numerous studies have shown that polyphenols can significantly impact aging hallmarks by regulating various cellular pathways and physiological processes. This section will elaborate in detail on the close relationship between polyphenols and the 12 hallmarks of aging, aiming to provide a comprehensive and systematic perspective for an in-depth understanding of the mechanism of action of polyphenols in anti-aging (Figure 3). Based on these mechanisms, it is critical to develop more dietary polyphenols as daily supplements for the human body, as they are expected to delay the aging process and promote healthy aging.

### 4.1. Polyphenols and Mitochondrial Dysfunction

Some evidence suggests that oxidative stress is the cause or result of mitochondrial dysfunction, which leads to cellular aging and altered function. Mitochondria are the major site of aerobic respiration in cells [137]. With aging, mitochondrial DNA (mtDNA) lesions, ROS accumulation, mitochondrial biogenesis, mitochondrial fission and fusion, dysregulation of mitochondrial dynamics, and dysregulation of mitochondrial protein homeostasis contribute to mitochondrial impairment, resulting in a spectrum of age-related diseases, including NDs and cancer [138]. Therefore, repairing mitochondrial damage may provide the possibility of delaying aging as well as treating age-related diseases. Polyphenols are rich in phenolic hydroxyl groups and have a strong ability to capture ROS, which enables polyphenols to act as mitochondrial protectors and exert anti-aging effects through antioxidation, free radical scavenging, balancing mitochondrial fusion and fission, and maintaining mitochondrial morphology and distribution. As a premium antioxidant, polyphenols available from food can contribute to the prevention of aging.

The accumulation of lesions and replication errors can lead to mutations in mtDNA, which can induce mitochondrial diseases [139]. The mutations of mtDNA accumulate highly in the cells of aging individuals, which may lead to mitochondrial dysfunction and age-related diseases. Intervening in the generation and accumulation of mtDNA mutations will be a potential approach for the prevention and treatment of age-related diseases. Quercetin, as a nutritional supplement, can raise the amount of mtDNA and improve mitochondrial biogenesis, thus enhancing physical performance [140]. It has been established that quercetin can repair damaged mitochondrial membrane potential and mtDNA damage, together with improved mitochondrial dynamics [141]. A study has also demonstrated that low doses of resveratrol can improve mitochondrial respiratory dysfunction in mtDNA mutant cells [142].

In response to impaired mitochondrial function, cells initiate a stress response mechanism for repairing mitochondrial function and balancing intracellular homeostasis, known as mitochondrial unfolded protein response (UPR^mt^). The UPR^mt^ is activated when mitochondria are slightly damaged, thus exerting a life-extending effect; when mitochondria are not damaged, the UPR^mt^ can also be induced by a non-self-dwelling manner; when mitochondria are severely damaged beyond repair, they are degraded or cleared by autophagy [143]. Mitochondrial input is impaired when mitochondria are under stress, allowing activating transcription factor associated with stress (ATFS-1) to be transported into the nucleus and activate UPR^mt^, which regulates transcriptional processes to achieve the re-establishment of dynamic mitochondrial homeostasis [144]. Knockdown of the mitochondrial electron transport chain (ETC) subunit cytochrome c oxidase-1 (cco-1) in neurons has been shown to induce activation of UPR^mt^, which subsequently exerts beneficial effects on the entire organism, including lifespan extension, through mitochondrial responses located in the intestine [145]. Collectively, the activation of UPR^mt^ is of critical importance for mitochondrial repair and even anti-aging research. As a dimethyl ether resveratrol analog, Pterostilbene is considered safe for human consumption. It was identified as a mitochondrial booster, which was found to be significantly enhanced in interaction with a mitochondrial cocktail (CoC3), which together increased the activity of sirtuins and activation of UPR^mt^, thus maintaining mitochondrial homeostasis and exerting neuroprotective, cardioprotective, and life-extending effects [146].

The fission and fusion of mitochondria are a dynamic process, regulated by multiple factors to adapt to the physiological needs of cells and environmental changes. When mitochondria suffer from DNA damage and protein aggregation, they may share healthy mitochondrial components through fusion. For example, resveratrol could improve cognitive impairment by increasing the size of hippocampal mitochondria through upregulating OPA1 and MFN2 proteins to promote mitochondrial fusion [147]. Furthermore, mitochondrial fusion can enhance mitochondrial function by diluting toxic superoxide, mutating mtDNA, and thus retarding the aging process of neuronal cells. Oxidative stress may lead to an increase in mitochondrial fission to remove the damaged mitochondrial parts. Mitochondrial fission is mainly regulated by dynamin-related protein 1 (DRP1), and rutin has been shown to repress mitochondrial fission. Cells treated with rutin after ethanol exposure showed reduced DRP1 expression, enhanced mitochondrial fusion, and an elongated morphology [148].

### 4.2. Polyphenols and Epigenetic Alterations

Epigenetics is a reversible genetic modification whose phenomena include DNA methylation, histone post-translational modification, and non-coding RNA [149,150]. Recent studies have reported that epigenetic changes are the main cause of aging in mammals, and that restoring epigenome integrity can reverse aging [151]. Dietary patterns, level of environmental pollution, lifestyle, and physical activity affect the molecular mechanisms of cells and the epigenome of cells and regulate gene expression, which in turn affects epigenetics and influences our life expectancy [149]. For the present, lifestyle changes and health service improvements are the main reasons for the increase in life expectancy in older adults.

DNA methylation elicits structural changes in chromatin and DNA, which is a relatively stable state of modification that can be hereditary to the next generation by the action of DNA methyltransferases. The alteration of DNA methylation is one of the major hallmarks of aging [152]. Reversing DNA methylation to slow overall aging may provide new insights into anti-aging research. Some studies have shown that GA has an inhibitory effect on DNA methylation and that GA has the ability to target aberrant epigenomes. DNMT1, a key enzyme involved in somatic cellular inheritance of DNA methylation, was significantly reduced in the nucleus or cytoplasm of GA-treated cells, and GA may be able to maintain the stability of the epigenome by inhibiting DNA methyltransferase (dnmt) [153]. GA-rich foods may exert anti-aging effects as an effective dietary intervention strategy.

Histones bind DNA into densely compressed chromatin and unbind parts of it if needed, and when they are unbundled, only then can they replicate, transcribe, and produce proteins. Accordingly, epigenetic factors can determine which genes are active or inactive in a given cell at a given time [151]. SIRT1 is an NAD^+^-dependent deacetylase. Resveratrol binds to SIRT1 at the amino-terminal site of the catalytic structural domain, increasing the affinity for acetylated substrates and constituting a metameric modification. Resveratrol has been proven to be an agonist of SIRT1. Resveratrol can increase AMPK activity, which in turn increases NAD^+^ concentration, leading to the activation of the SIRT1 pathway, thus exerting anti-aging effects [154].

Non-coding RNA includes snRNA, snoRNA, and microRNA (miRNA), of which miRNA is a type of evolutionarily conserved non-coding small-molecule RNA that regulates gene expression at the translational level. Quercetin has a wide range of biological activities and is also highly beneficial to human health from an epigenetic point of view. In different types of cancers, quercetin can regulate microRNAs. For example, it can increase the expression of miR-146a in breast cancer cells to inhibit tumor growth, and it can also regulate miR-146a in lung adenocarcinoma by affecting the *K-Ras* pathway [155]. Quercetin may also ameliorate cognitive dysfunction in AD animal models by reducing amyloid β-protein, tau phosphorylation, neuroinflammation, or modulating miRNAs such as miR-26a, miR-125b, and miR-132 [156]. After quercetin administration, liver-specific levels of miRNA miR-122, which is involved in lipid metabolism, promoting tumor cell apoptosis and inhibiting oncogene expression in the liver, were significantly elevated [157]. In addition, long-term quercetin supplementation increases miRNA expression in the hippocampus, ameliorates cognitive impairment, and slows the aging process [158].

### 4.3. Polyphenols and Disabled Macroautophagy

There are three main types of autophagy: macroautophagy, microautophagy, and molecular chaperone-mediated autophagy, where macroautophagy is the main form of autophagy. During autophagy, cytoplasmic material is segregated in double-membrane vesicles to form autophagosomes, fusing with lysosomes to digest luminal contents [159]. Autophagy plays an important role in intracellular homeostasis and is a self-regulatory process that arises in response to nutrient limitation, hypoxia, and endoplasmic reticulum stress [160]. Autophagy proactively degrades misfolded proteins or aggregates, removes superfluous, damaged, and harmful organelles, and maintains certain protein or lipid levels to safeguard normal cellular function [161]. Compromised autophagy is one of the newly added hallmarks of aging. Autophagy capacity decreases with aging, resulting in slower organelle renewal and metabolism and a decline in the body’s repair capacity, which predisposes the induction of various age-related diseases. Autophagy assumes a pivotal role in the therapy of NDs. Aberrant protein aggregation represents a prevalent pathological hallmark across various neurodegenerative disorders, and the pharmacological modulation of autophagy to enhance the elimination of misfolded proteins and other neurotoxic agents holds promising therapeutic implications for the management of NDs [162]. Polyphenols, as an accessible natural substance, can act as an autophagy inducer to promote the apoptosis of senescent cells, thereby exerting anti-aging effects.

α-Synuclein (α-Syn) is a soluble protein expressed presynaptically and perinuclear in the central nervous system, which is the main component of Lewy bodies. The aggregation of α-Syn and the formation of Lewy bodies are associated with neuronal death, both of which are pathologic hallmarks of PD [163]. Copper, as a trace metal, can promote the aggregation and misfolding of α-Syn protein, leading to the development of PD. Quercetin has been found to regulate the expression of endogenous α-Syn protein and enhance chaperone-mediated autophagy. Additionally, it can mitigate copper-induced apoptosis and endoplasmic reticulum stress in SH-SY5Y cells through autophagy [164]. These findings hold great promise for utilizing quercetin as a potential therapeutic agent for PD treatment.

Aβ is formed by the proteolytic cleavage of β-amyloid precursor protein by β and γ secretase, which is highly neurotoxic after precipitation and aggregation within the cell matrix. Aberrant accumulation of Aβ represents a prominent predisposing factor for AD [165]. Targeting the elimination of Aβ deposition can partially mitigate cognitive and functional decline, thereby delaying the onset of AD. Polyphenols in lychee seed exhibit inhibitory effects on Aβ (25–35)-induced NLRP3 inflammasome activation through the AMPK/mTOR/ULK1-mediated autophagy pathway, thereby safeguarding the integrity of the blood–brain barrier in AD [166].

Both AMPK and mTOR are classical inducers of autophagic responses during stress and exercise. EGCG is a polyphenol antioxidant found in a variety of teas with free radical scavenging properties, which can be replenished by simply drinking tea. By regulating the equilibrium of the mTOR–AMPK pathway upon endoplasmic reticulum stress, EGCG might encourage autophagy-dependent survival and boost cell viability [167]. As an autophagy inducer, resveratrol expands longevity. It mitigates aging symptoms through SIRT1-dependent induction of autophagic and non-autophagic pathways [34]. On the other hand, following resveratrol therapy, p38 was activated and enhanced autophagic flux in a dose-dependent manner, which could increase autophagy induced by the AMPK/p38MAPK pathway and exert a protective effect [168]. In conclusion, dietary polyphenol intake has some anti-aging effects and lowers the etiology of neuropathy and neurodegenerative disorders through autophagy.

### 4.4. Polyphenols and Deregulated Nutrient-Sensing

Nutrient sensing is the ability of a cell to perceive and react to energy substrates such as glucose, fatty acids, and ketones. Once these substrates are recognized, the cell affects gene expression and other signaling pathways for physiological processes. The insulin-like growth factor (IGF-1) signaling (IIS) pathway, the mTOR signaling pathway, and the AMPK pathway are the three pivotal pathways for nutrient sensing [5]. The dysregulation of nutrient sensing can lead to age-related and metabolic illnesses. Dietary restriction (DR) and calorie restriction (CR), which is the reduction in food intake without causing malnutrition, can mitigate many of the deleterious effects of aging and may be a promising way to improve health and longevity in humans [169,170]. A recent study utilizing the epigenomic DunedinPACE algorithm discovered that a 25% reduction in calorie intake slowed the rate of aging in healthy adults by 2% to 3%, which further supports the potential of CR in anti-aging [171]. Multiple genes in the cell that are sensitive to nutritional deficiency can be activated by CR, which causes the cell to stop growing and initiate the process of defending or repairing itself [172]. This causes the body’s antioxidant capacity to increase while the inflammatory response decreases. The production of the aging-associated protein Secreted Protein Acidic and Rich in Cysteine (SPARC), which prevents detrimental inflammation and enhances health in the elderly, is reduced by moderate CR [173]. Some polyphenols have the effect of calorie restriction mimics and can exert anti-aging effects by regulating nutrient sensing.

Resveratrol is a natural polyphenol that affects energy metabolism and mitochondrial function. Resveratrol was found to mimic CR in yeast as early as 20 years ago by increasing Sir2 expression, thereby enhancing DNA stability and extending lifespan by 70% [65]. It has been clinically proven that resveratrol supplementation can mimic the effects of CR by inducing metabolic changes in obese populations [174]. Researchers discovered that CR decreased the age-related SASP markers PAI-1 and MMP-9 and increased the anti-aging effects of dasatinib in combination with quercetin in middle-aged cynomolgus macaques [97].

In summary, polyphenols can exert the effects of DR by modulating metabolism, while moderate DR activates the AMPK pathway and increases inhibition of the rapamycin complex 1 (TORC1) pathway, increasing autophagy, degrading misfolded proteins and depleted organelles to maintain cellular homeostasis, and extending model organism lifespan. The AMPK pathway’s activation also encourages the action of the SIRT1 and FOXO factors, which help the cell withstand famine, nutritional shortages, and stress responses, thus expanding its lifespan.

### 4.5. Polyphenols and Chronic Inflammation

Age-related diseases such as osteoarthritis, atherosclerosis, and neuroinflammation are all accompanied by chronic inflammation, which indicates that chronic inflammation plays an important role in aging. Early studies showed that the cells of aged mice would exhibit a rejuvenated state when placed in the serum of young mice, while the cells of young mice would lose vitality in the serum of aged mice [175]. Further research demonstrated that the serum of young mice contained lower levels of inflammatory factors and presented a more favorable growth microenvironment. This phenomenon suggested that the key determinant of cell vitality was not the age of the cells but chronic inflammation. The factors secreted by senescent cells, including pro-inflammatory cytokines and chemokines, growth factors, angiogenic factors, and matrix metalloproteinases, are known as the SASP, which can promote chronic inflammation and induce senescence in normal cells. Failure to clear senescent cells and SASP will result in a vicious cycle of inflammation and aging [176].

The chronic inflammation resulting from the imbalance between the accumulation and clearance of ROS serves as a pathogenic factor for osteoarthritis (OA). Polyphenolic compounds exhibit anti-inflammatory and antioxidant activities and can suppress oxidative stress and inflammation in OA joints by suppressing the pro-inflammatory signaling pathways such as MAPK, AP1, and NF-κB [177]. Studies have shown that oral administration of quercetin may be able to inhibit the generation of ROS by activating the AMPK/SIRT signaling pathway to reduce inflammatory factors such as NO, MMP-3, and MMP-13, thereby improving the symptoms of OA rats [140]. Curcumin can effectively alleviate the pain caused by OA, protect against IL-1β-induced chondrocyte apoptosis, and inhibit the production of cPLA2, COX-2, and 5-LOX [178].

Neuroinflammation is the activation of glial cells caused by infections, NDs, or autoimmune diseases, which leads to the release of inflammatory mediators and causes damage to neurons. In the aging brain, the dysfunction of microglia and glial cells can lead to the persistence of chronic inflammation. Pomegranate (PU) has a strong antioxidant capacity and is widely found in pomegranates, raspberries, blueberries, and chestnuts. Studies have found that in accelerated aging and natural aging mouse models, PU not only reduces the activation of microglia and the proliferation of astrocytes, lowers the levels of MDA and ROS, but also inhibits the activation of NLRP3 inflammasomes, reduces oxidative stress responses, and related inflammatory cytokines such as IL-6, TNF-α, IL-18, and IL-1β [179]. Thus, PU could improve learning and memory deficits and prevent neuroinflammation.

### 4.6. Polyphenols and Genomic Instability

Genomic instability is driven by many endogenous and exogenous damages, resulting in increasing genomic alterations, involving damage to nuclear DNA (nDNA) and mtDNA, as well as changes in the structure and stability of chromosomes. Once DNA damage is not repaired in time, it will lead to incorrect genetic information transmission, interfere with the normal physiological functions of cells, and accelerate the aging process. Polyphenols primarily repair DNA damage through their antioxidant capacity, preserve genomic stability, and thus show anti-aging properties.

ROS has been demonstrated to be involved in regulating cell signal transduction and apoptosis and is one of the potential causes of DNA damage. EGCG could prevent CuO nanoparticle/H_2_O_2_-mediated DNA damage at very high concentrations due to its antioxidant behavior [180]. Radiation can also cause DNA damage. Studies have shown that pretreatment with 0.5 mM resveratrol before radiation can significantly reduce DNA damage, while the addition of 0.5 mM resveratrol during or after radiation is more effective in alleviating DNA damage caused by γ-rays [181].

ATM (ataxia telangiectasia mutated) and ATR (ATM and Rad3-related) kinases are crucial components of the intracellular DNA damage response mechanism. ATM responds to DNA double-strand breaks, while ATR responds to DNA single-strand damage and replication issues [182]. Once activated, they can suspend the cell cycle to gain time for repair and participate in the repair process. Research has indicated that the metabolites of procyanidins could alter the DNA damage response and promote the ATR–Chk1 and ATM–Chk2 pathways in vitro, reducing DNA damage in human lung epithelial cells and fetal hepatocytes induced by chemical carcinogens [183].

### 4.7. Polyphenols and Dysbiosis

The human gut is a highly diverse and dynamic micro-ecosystem with approximately 100 trillion microorganisms, and the species of which amount to several thousands. The aging process is typically accompanied by substantial alterations in the structure, abundance, and activity of the gut microbiota. Some animal studies have transferred the microbiota of feces from young animals to aged animals and found that the young microbiota reversed the manifestations related to aging [184,185,186]. Therefore, the gut microbiota is extensively involved in the processes of disease occurrence and development as well as aging by regulating the metabolic processes [187], inflammatory responses [188], and microbiota–gut–brain axis [189] of the organism.

Apart from directly consuming products containing probiotics, some specific foods have been scientifically proven to be capable of improving the gut microbiota. For instance, the Mediterranean diet, widely recognized as a healthy dietary pattern and is considered conducive to longevity, can significantly enhance the gut microbiota of the elderly and thereby improve their overall health status [190]. The Mediterranean diet provides a considerable amount of natural beneficial substances, among which dietary polyphenols constitute an important component. Oral administration of green tea polyphenol EGCG has been scientifically demonstrated to regulate the gut microbiota effectively, significantly inhibit the abundance of pathogenic bacteria, robustly enhance the integrity of the colonic barrier, and effectively alleviate colonic inflammation, which is highly beneficial to human health and simultaneously exhibits its considerable potential in anti-aging [52,191]. Furthermore, when chlorogenic acid and EGCG are used in combination, they can significantly enhance the anti-inflammatory and antioxidant effects, effectively inhibit the accumulation of ROS mediated by D-galactose, optimize the ratio of gut microbiota, maintain the diversity of microorganisms, and thereby alleviate intestinal aging [192]. The senolytic drug combination, Dasatinib plus Quercetin, can significantly reduce the contents of senescent cells and inflammatory factors in the small intestine and large intestine of mice and simultaneously regulate specific microbiota characteristics, thereby delaying intestinal inflammation and aging in mice [9]. Quercetin can also increase the abundance of beneficial microbiota in mice with pulmonary fibrosis, improve their intestinal flora imbalance, and delay the aging of alveolar epithelial cells [193].

Furthermore, certain dietary polyphenols mainly exert their anti-aging effects through their intestinal metabolites. UA is the metabolite of EA in the gut. Differences in individual ability to produce UA may cause significant individual variations in the health effects of EA. Studies have shown that in rat models with high UA production, administration of EA can improve memory and cognitive dysfunction while an increase in beneficial bacteria in the gut microbiota was observed [70]. Demethylcurcumin and bisdemethoxycurcumin are metabolites of curcumin in the gut, which have a neuroprotective effect on AD mice [48].

### 4.8. Polyphenols and Other Hallmarks of Aging

In addition to the above seven hallmarks of aging, polyphenols also play significant roles in altered cellular communication, stem cell exhaustion, cellular senescence, telomere attrition, and loss of proteostasis.

For instance, quercetin possesses remarkable antioxidant properties and can effectively reduce cellular oxidative stress damage, thereby maintaining normal physiological functions and communication capabilities of cells. Meanwhile, quercetin can also significantly decrease the release of SASPs, thereby regulating the communication between immune cells and maintaining the balance of the immune system [95]. EA could increase the proliferation of neural stem cells through the Wnt/β-catenin signaling pathway to assist in the treatment of nerve dysfunction and aging [72]. Rutin effectively delays aging by inhibiting the expression of the full-spectrum SASP and precisely targeting senescent cells [121]. Cy3G, as a natural inhibitor of CD38, can moderately increase the expression of telomerase reverse transcriptase, which is conducive to restoring the function and vitality of cells and extending the lifespan [109]. Impaired proteostasis is closely related to protein-conformational diseases like AD, PD, and prion diseases. Research indicates that EA can not only inhibit the formation but also disassemble the fibers of the prion protein, thus making it a promising candidate for the therapy of prion diseases [194].

In conclusion, numerous studies have shown that polyphenols have exerted significant benefits in every hallmark of aging. Whether it is the aging of cells, the deterioration of tissues, or the decline of the overall function of the organism, polyphenols have demonstrated positive and important roles.

## 5. Conclusions

With the hope that dietary polyphenols will eventually be utilized in place of pharmaceuticals to promote health, interest in polyphenol-rich natural products such as dietary supplements have increased significantly in recent years. Increasing evidence suggests that polyphenolic compounds such as EA, GA, rutin, quercetin, fisetin, and anthocyanins may promote health and extend lifespan in a variety of animal models through several mechanisms, genomic instability, telomere attrition, epigenetic alterations, loss of proteostasis, disabled macroautophagy, deregulated nutrient-sensing, mitochondrial dysfunction, cellular senescence, stem cell exhaustion, altered intercellular communication, chronic inflammation, and dysbiosis. Aging and age-related diseases are intricate. This review synthesizes the 12 hallmarks of aging that have been newly put forward, highlighting the new evidence of anti-aging related to polyphenols. Aging is intrinsic and demands to be comprehended as a whole. Currently, studies typically commence from one or several hallmarks of aging. Regarding the anti-aging effect of polyphenols, each hallmark of aging should be regarded as a point of entry for future exploration of the aging process and the development of new anti-aging drugs. Furthermore, although various polyphenols have been proven to prevent or treat age-related diseases in various model organisms based on studies, there are still many vacancies in the aspect of clinical research.

## 6. Outstanding Questions

(1)What factors cause individual differences in the metabolism and bioavailability of polyphenols in the body? How can the intake of polyphenols be optimized to achieve the best anti-aging effect?(2)What are the synergy mechanisms and dynamic regulation of the anti-aging effect of polyphenols?(3)Regarding the effects of polyphenols in the human body, what is their long-term efficacy like? And what conclusions have been drawn from the relevant safety clinical trials?(4)The aging process involves the interaction of multiple organs and systems, but existing studies mainly focus on individual hallmarks of aging. What is the comprehensive impact of polyphenols on the aging of multiple organ systems throughout the body?

## Figures and Tables

**Figure 1 nutrients-16-03305-f001:**
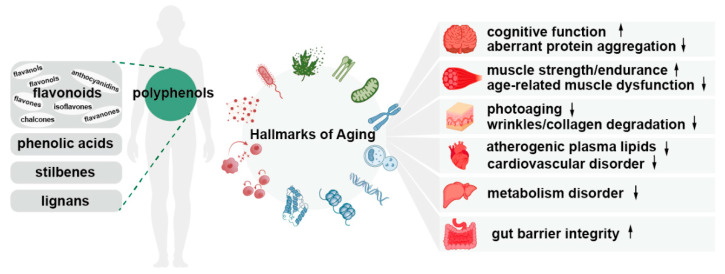
The anti-aging effects of polyphenols on different organs and tissues. Polyphenols have significant biological effects on different tissues and organs as indicated by in vivo preclinical studies and clinical trial data. They boost physiological functions that decline with age and lower the occurrence of age-associated pathologies. Upward arrows stand for up-regulation. Downward arrows stand for down-regulation.

**Figure 2 nutrients-16-03305-f002:**
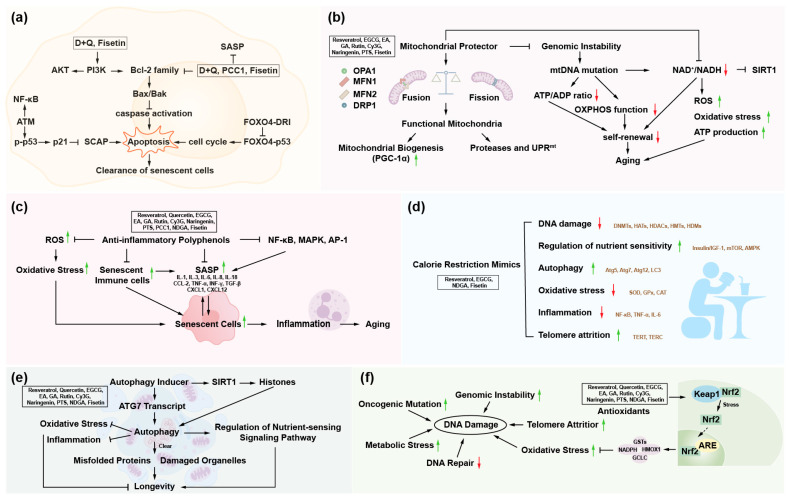
Polyphenols exhibit anti-aging effects as senolytics, mitochondrial protectors, anti-inflammatory agents, calorie restriction mimics, autophagy inducers, and antioxidants. (**a**) Cellular senescence and senolytics. (**b**) Mitochondrial dysfunction and mitochondrial protector. (**c**) Chronic inflammation and anti-inflammatory polyphenols. (**d**) Dysregulated nutrient-sensing and calorie restriction mimics. (**e**) Disabled macroautophagy and autophagy inducer. (**f**) Genomic instability and antioxidants. Red arrows stand for down-regulation. Green arrows stand for up-regulation. Abbreviations: AKT, protein kinase B; AP-1, activator protein 1; ATG12, autophagy-related 12; ATG5, autophagy-related 5; ATG7, autophagy-related 7; ARE, antioxidant response element; ATM, ataxia telangiectasia mutated; Bak, Bcl-2 homologous antagonist/killer; Bax, Bcl-2-associated X protein; Bcl-2, B-cell lymphoma 2; CCL-2, C-C motif chemokine ligand 2; CAT, catalase; Cy3G, cyanidin 3-glucoside; DNMT, DNA methyltransferase; DRP1, dynamin-related protein 1; EA, ellagic acid; EGCG, epigallocatechin-3-gallate; FOXO4, fork head box O transcription factor 4; FOXO4-DRI, fork head box O transcription factor 4-D-retro-inverso; GA, gallic acid; GCLC, glutamate-cysteine ligase catalytic subunit; GST, glutathione S-transferase; HAT, histone acetyltransferase; HDAC, histone deacetylase; HDM, histone demethylase; HMOX1, heme oxygenase 1; IGF-1, insulin-like growth factor 1; IL-1, interleukin-1; IL-18, interleukin-18; IL-3, interleukin-3; IL-6, interleukin-6; IL-8, interleukin-8; Keap1, Kelch-like ECH-associated protein 1; LC3, microtubule-associated protein 1 light chain 3; MAPK, mitogen-activated protein kinase; MFN1, mitofusin 1; MFN2, mitofusin 2; NDGA, nordihydroguaiaretic acid; NF-κB, nuclear factor kappa-B; NAD^+^, nicotinamide adenine dinucleotide; NADH, nicotinamide adenine dinucleotide hydrogen; NADPH, nicotinamide adenine dinucleotide phosphate; Nrf2, nuclear factor erythroid 2-related factor 2; OPA1, optic atrophy 1; OXPHOS, oxidative phosphorylation; PGC-1α, peroxisome proliferator-activated receptor gamma coactivator 1-alpha; PI3K, phosphatidylinositol 3-kinase; PCC1, procyanidin C1; PTS, pterostilbene; ROS, reactive oxygen species; SASP, senescence-associated secretory phenotype; SIRT1, sirtuin 1; SOD, superoxide dismutase; TGF-β, transforming growth factor-beta; TERC, telomerase RNA component; TERT, telomerase reverse transcriptase; TNF-α, tumor necrosis factor-alpha; UPR^mt^, mitochondrial unfolded protein response; mTOR, mammalian target of rapamycin.

**Figure 3 nutrients-16-03305-f003:**
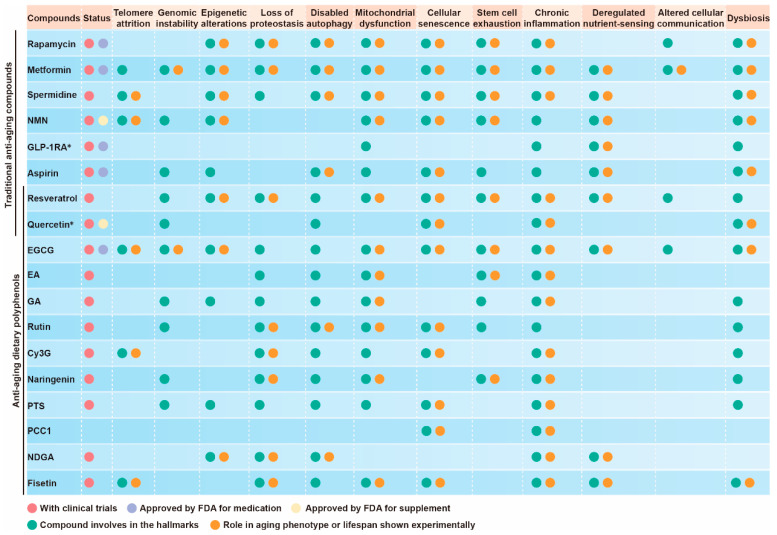
The role of anti-aging treatments, including polyphenols in the 12 hallmarks of aging. Explanation: Quercetin* itself is a natural polyphenolic compound. However, the senolytic drug combination group “quercetin + dasatinib” is more worthy of attention due to its anti-aging effect. Therefore, quercetin in this table refers to this drug combination group. NMN was originally a dietary supplement but has now been withdrawn by the FDA. GLP-1RA* refers to GLP-1 receptor agonists, widely used in obesity and diabetes. The latest research evidence indicates that it has potential anti-aging effects. “With clinical trials” means that there are clinical trial records of the compound on the official clinical trials website. “Approved by FDA for medication” means whether the FDA has approved this compound as a drug. “Approved by FDA for supplement” means whether the FDA has approved this compound as a dietary supplement. “Compound involvement in the hallmarks” and “role in aging phenotype” mainly refer to whether the polyphenol inhibits the 12 hallmarks or regulates the hallmarks to have a beneficial effect on aging experimentally (or in laboratory studies) before clinical trials (https://clinicaltrials.gov/) (accessed on 17 September 2024).

**Table 1 nutrients-16-03305-t001:** Dietary anti-aging polyphenols and potential mechanisms.

Polyphenols	Source	Age-Related Conditions	Model	Dosage	Anti-Aging Activity and Proposed Anti-Aging Mechanism
Resveratrol 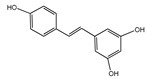	Grapes, red wine, peanuts, and blueberries	Age-related motor deficits	Older adults	500 mg/day, 1000 mg/day	A pilot randomized controlled trial indicates that the combined application of exercise training and resveratrol in elderly people with functional limitations could improve skeletal muscle mitochondrial function and exercise-related physical function indicators [58].
		Sarcopenic obesity	Sprague-Dawley (SD) rats	0.4% of diet	Resveratrol could ameliorate mitochondrial dysfunction and oxidative stress, thereby improving protein metabolism and helping to prevent sarcopenic obesity in the elderly [59].
		Aging	ddY mice (derived from a non-inbred strain carrying a retrovirus causing significant mortality with age)	0.4 g/kg of diet	Resveratrol could reduce the level of acetylated proteins in muscles and restore autophagic activity, thereby alleviating age-related sarcopenia and cardiomyocyte hypertrophy [60].
		Aging	HtrA2 KO mice (the absence of serine/threonine protease HtrA2 causes a PD phenotype)	25 mg/kg body weight (BW)	Resveratrol treatment can extend the lifespan of HtrA2 KO mice and delay the deterioration of the motor phenotype by attenuating apoptosis at the level of Bax [61].
		Aging	*Caenorhabditis elegans* (*C. elegans*)	100 mM	Resveratrol, as a Sirtuin activator, can simulate calorie restriction and extend the lifespan of *C. elegans* [62].
		Aging	*Drosophila melanogaster*	200 μM	Supplementation of resveratrol to larval can effectively eliminate ROS, thereby extending the adult lifespan of fruit flies and not reducing their reproductive capacity [63].
		Ovarian aging	*N. guentheri*	200 μg/g food	Resveratrol could delay ovarian aging by alleviating inflammation and ER stress through the SIRT1/NRF2 pathway [64].
		Aging	*S. cerevisiae*	10, 100, 500 μM	Resveratrol could mimic calorie restriction by stimulating Sir2, increase DNA stability, and extend lifespan by 70% in yeast [65].
		Aging	Human mononuclear cells (PBMCs)	5 µM	Compared with the elderly, resveratrol exerts better antioxidant and anti-inflammatory effects in PBMCs of middle-aged individuals [66].
		Brain aging	Hippocampal astrocytes	10 μM	Resveratrol could increase the antioxidant defense capacity and reduce pro-inflammatory cytokines in hippocampal astrocyte cultures of rats of all ages, improving NDs [67].
		Age-related motor deficits	SH-SY5Y cells	1 or 5 μM	Resveratrol alleviates age-related motor deficits by promoting the survival of dopaminergic neurons and the activation of the ERK1/2 pathway [68].
Ellagic acid 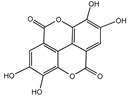	Berries, nuts, tea, and medicinal plants [69]	Aging	D-gal-induced aging rats	120 mg/kg BW	EA improved oxidative damage and inflammation in D-gal-induced aging rats [70].
		Hepatic lipid metabolism disorders induced by aging	25-month-old rats	30 mg/kg BW	EA affected lipid metabolism in the aged liver via the sirt1/AMPK/sreBP-1c/PPAr-α pathway for age-related metabolic disorders [71].
		Brain aging	SD rats	10, 30, 90 mg/kg BW	EA could increase the proliferation of brain neural stem cells and protect brain cell activity through the Wnt/β-catenin signaling pathway to help treat nerve dysfunction, NDs, and aging [72].
		Osteoporosis	C57BL/6 mice, bone mesenchymal stem cells	Mice: 10, 50 mg/kg BW; cell: 0 μM–30 μM	EA could promote bone formation by activating the SMAD2/3 signaling pathway to ameliorate aging-induced osteoporosis [73].
		Skin aging	Male SKH-1 hairless mice, human dermal fibroblasts	Mice: 10 μmol/L, cell: 1–10 μmol/L	EA could reduce wrinkles and UV-induced skin inflammation to improve photoaging by decreasing UV-B-induced collagen degradation and inflammatory reactions [74].
		Aging	*C. elegans*	50 μM	EA reduced the injury caused by UV radiation and enhanced stress resistance to extend the lifespan of *C. elegans* [75].
		Aging	*Drosophila melanogaster*	100 μM and 200 μM	EA can up-regulate the expression of dFOXO, CAT, and SOD, thereby extending lifespan [76].
		Aging	SH-SY5Y cells	0.1–1 μM	Lower concentrations of EA could provide better anti-aging benefits than higher concentrations of EA and metformin, presumably via the PPAR-γ/ HO-1 signaling pathway [77].
		Skin sagging and wrinkling	Human dermal fibroblasts	2 μg/mL	EA improved skin extracellular matrix production of elastin and collagen and improved skin fine wrinkles [78].
Urolithin A 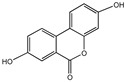	Gut microbiota metabolite of EA	Aging	Healthy elderly male and female	250, 500, 1000 and 2000 mg	UA could promote mitochondrial autophagy and improve muscle health in old animals and in preclinical models of aging [79].
		Aging	D-gal-induced aging mice	3, 15 mg/kg BW	Uro-A from the colon can prevent D-gal-induced aging in mice by blocking NF-κB and mTOR targets, improving motor and cognitive abilities [80].
		Alzheimer’s disease (AD)	3xTg-AD mice, B6129SF2/J mice	5 mg/kg BW	UA could improve the cognition of 3xTg-AD mice and prolong longevity in normal aging mice by inducing autophagy and increasing amyloid-β (Aβ) clearance in neuronal cells [81].
		Aging	*C. elegans*, C57BL/6 mice	*C. elegans*: 50 μM, mice: 50 mg/kg BW	Urolithin A could induce mitophagy and improve mitochondrial and muscle function in *C. elegans* and rodents, leading to life extension [82].
		Brain aging	H_2_O_2_-induced PC12 cell, D-gal-induced aging mice	Cell: 10, 30, 50 μg/mL, Mice: 50, 100, 150 mg/kg body weight	Urolithin A can exert neuroprotective effects and delay brain aging by activating miR-34a-mediated SIRT1/mTOR signaling pathway [83].
		Aging	D-gal-induced aging mice	50, 100, 150 mg/kg BW	UA could attenuate D-gal-induced liver injury in aged mice via antioxidant, anti-inflammatory, and anti-apoptotic properties [84].
Gallic acid 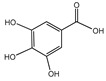	*Rheum palmatum*, *Eucalyptus robusta*, *Cornus officinalis*	Age-associated thymic involution	D-gal-treated mice	200, 250, 500 mg/kg body weight	GA administration may ameliorate age-related thymic degeneration and enhance immune function in the elderly by stimulating FoxN1 expression, increasing proliferating cells, and decreasing apoptotic cells [85].
		Aging	H_2_O_2_-induced rat’s embryonic fibroblast cells	1000 μM	GA could postpone aging through its antioxidative stress potential and modulation of mitochondrial complexes’ activities [86].
		AD	APP/PS1 transgenic AD mouse model	20 mg/kg body weight	EA could inhibit neuroinflammation and stabilize brain oxidative stress [87].
Rutin 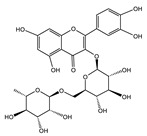	*Fagopyrum esculentum*, *Ruta graveolens*, *and Sophora japonica* [88]	Aging	*Drosophila melanogaster*	200 μM and 400 μM	Rutin could improve the resistance of male and female *Drosophila melanogaster* fed with a high-fat diet, increase the expression of age-related genes, and extend longevity [89].
		Huntington’s disease	*C. elegans*	15, 30, 60, 120 μM	Rutin could inhibit polyglutamine protein aggregation in muscle, decrease neuronal death, and extend longevity through antioxidant, autophagy, and insulin/IGF1 signaling pathways [90].
		Aging	D-gal-treated mice	50 mg/kg body weight	Rutin could enhance the biochemical indicators of aging rats by exerting antioxidant effects and regulating apoptosis-related proteins to inhibit cell apoptosis [91].
		Age-related metabolic dysfunction	Twenty-month-old rats	25, 50 mg/kg body weight	Rutin suppresses aging-associated mitochondrial dysfunction, endoplasmic reticulum stress, and oxidative stress, thus improving the response to age-related metabolic dysfunction [92].
		AD	TgAPP mice (a model for AD)	30 mg/kg body weight	Rutin could raise GSH/GSSG levels, reduce MDA levels, and inhibit APP expression and BACE1 activity, resulting in anti-AD and anti-aging benefits [93].
Quercetin 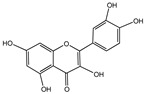	Grapes, peaches, onions, garlic [94]	Age-related metabolic dysfunction	Twenty-one-month-old mice	Dasatinib (5 mg/kg BW) and quercetin (50 mg/kg BW)	Dasatinib + quercetin (D + Q) lowers inflammation in adipose tissue and enhances systemic metabolic function in older people [95].
		Age-dependent progression of disc degeneration	C57BL/6 mice	Dasatinib (5 mg/kg BW) and quercetin (50 mg/kg BW)	D + Q can target senescent cells non-invasively and lessen the impact of age-dependent degeneration [96].
		Aging	*Macaca fascicularis*	Dasatinib (5 mg/kg BW) and quercetin (50 mg/kg BW)	D + Q could prevent aging by strengthening the gut barrier, boosting immunity, and combating inflammation [97].
		Aging	C57BL/6 mice	Dasatinib (5 mg/kg BW) and quercetin (50 mg/kg BW)	D + Q could lead to selective elimination of senescent cells and the release of senescence-associated pro-inflammatory cytokines [98].
		Intestinal senescence	BALB/c mice	Dasatinib (5 mg/kg BW) and quercetin (50 mg/kg BW)	Long-term D + Q treatment could decrease the gene expression of senescence and inflammation to alleviate intestinal senescence [9].
Naringenin 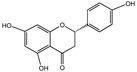	*Ribes meyeri*	Aging	C57BL/6 mice	25, 50, 100 mg/kg BW	Naringenin could reduce neuronal damage and mitigate systemic inflammation induced by LPS by modulating the expression of NF-κB/TNFα/COX-2/iNOS/TLR4/GFAP [99].
		Retinal degeneration	C57BL/6 mice	100 mg/kg BW	Oral administration of naringenin could regulate mitochondrial dynamics and autophagy to counteract aging-related retinal degeneration [100].
		Aging	*C. elegans*, 6-month-old C57BL/6J mice	*C. elegans*: 100 μM, mice: 100 mg/kg BW	Naringenin could extend the lifespan of *C. elegans* and slow brain aging in mice by increasing the expression of SIRT enzymes, promoting the activity of metabolic enzymes, and upregulating the expression of anti-aging markers [101].
Cyanidin-3-O-glucoside (Cy3G) 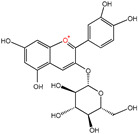	Vegetables and berries [102]	AD	HMC3 cell, APPswe/PS1ΔE9 mice (a model for AD)	cell: 50 μM, mice: 30 mg/kg BW	Cy3G could eliminate accumulated β-amyloid and modulate microglial polarization by activating PPARγ and promoting Aβ42 phagocytosis through TREM2 overexpression [103].
		AD	APPswe/PS1ΔE9 mice	30 mg/kg BW	Cy3G could reduce lysosome-associated protein expression, increase autophagy, and modulate the PI3K/Akt/GSK3β signaling pathway to protect neurons and enhance cognitive performance in AD mice [104].
		AD	SD rats received Aβ in the hippocampus	10 mg/kg BW	Cy3G could attenuate Aβ-induced tau protein hyperphosphorylation and GSK-3β hyperactivation, possibly rescuing Aβ-induced cognitive deficits through GSK-3β/tau variation [105].
		AD	APPswe/PS1ΔE9 mice	30 mg/kg BW	Cy3G may exert therapeutic effects on AD through antioxidant and immunomodulatory mechanisms [106].
		Parkinson’s disease (PD)	MPTP-induced C57BL/6J	10, 20, 40 mg/kg BW	Cy3G could play a role in the treatment of PD by modulating the structure and metabolism of gut microbiota [107].
		Aging	*C. elegans*	12.5, 25, 50 μg/mL	Cy3G could enhance resistance and extend the lifespan of polystyrene-exposed *C. elegans* through the DAF-16 pathway [108].
		Aging	H9c2 cells	1 mM	Cy3G could decrease CD38 expression, increase Sirt6 expression in tissues, and restore NAD^+^ and NK cell levels to exert anti-aging effects through CD38-Sirt6 signaling [109].
Procyanidin C1 (PCC1) 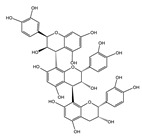	Grape seed extract	Aging	PSC27 cells, WI38 cells, HUVEC cells; C57BL/6J mice	Cell: 1, 5, 10 μg/mL Mice: 20 mg/kg BW	Low concentrations of PCC1 can inhibit the formation of SASP, while at higher concentrations, it may selectively kill senescent cells by promoting the production of ROS and mitochondrial dysfunction [110].
		Aged retina	Aged mice	8 mg/kg of diet	Long-term PCC1 treatment could relieve function and structural impairment in the aged retina and reduce the accumulation of senescent cells and secretion of SASP [111].
Nordihydroguaiaretic acid 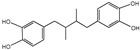	*Larrea tridentata*	Aging	UM-HET3 mice (designed to model the aging process and age-related diseases)	2.5, 5 g/kg of diet	Nordihydroguaiaretic acid could increase the survival rate of male mice but does not change the survival rate of female mice, which might be explained by gender differences in steady-state levels or drug metabolism [112].
Pterostilbene 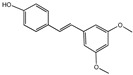	Blueberries	AD	Senescence accelerated mouse prone 8 (SAMP8) mice	120 mg/kg of diet	Pterostilbene could increase the expression of peroxisome proliferator-activated receptor (PPAR) α, improve cognitive function and cellular stress regulation ability, and alleviate AD symptoms [113].
		AD	18-month-old rats	22.5 mg/kg BW	Pterostilbene could increase the expression of postsynaptic density protein 95 and improve the cognitive ability of elderly rats with mild cognitive impairment AD [114].
		AD	19-month-old male Fischer	40, 160 mg/kg of diet	Pterostilbene can effectively reverse cognitive behavioral deficits and dopamine release and improve age-related cognitive degeneration [115].

## Data Availability

No data availability.

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
