# Peer review of "Dietary Polyphenols as Anti-Aging Agents: Targeting the Hallmarks of Aging"

_nutrients, 2024, doi:10.3390/nu16193305_

Round 1

Reviewer 1 Report

Comments and Suggestions for Authors

Dietary Polyphenols as Anti-Aging Agents: Targeting the Hall- 2 marks of Aging. Overall, the manuscript addresses important questions in a biological process inherent to life. The proposal of mechanisms underlying the regulated events and the visual aids are the strengths of the manuscript. Some aspects are suggested for improvement at the authors' consideration.

Abstract

The abstract is well structured, informative, and provides an accurate description of the potential benefits of polyphenols on aging.

Introduction

Lines 26-30: In the introduction, certain links could be replaced with references to improve the reading flow.

In line 33, it states: "aging is an unavoidable degradative disease" – Is it considered a disease, or can it be referred to as a biological process inherent to life?

The statements in lines 70-75 regarding the potential against dyslipidemias are not necessarily essential for this justification.

The statements from lines 99 to 103 are also too specific to cardiovascular or hepatic activities and not necessarily associated with aging. This information detracts from the focus on the central topic, and although these are events related to aging, they are very specific and presented in an isolated manner.

Line 160: the expression "in conclusion" is not appropriate for this section.

Figure 1 is very well-made and highlights many points that are not discussed in depth. The hallmarks of aging could be explored further.

Table 1: Is it possible to exchange "disease" for "condition"? I may be mistaken, but I insist on the question of whether aging is a disease.

Most of the polyphenols in Table 1 have an acronym. It would be advisable to standardize all of them with the same format. If this is the case, avoid repeating the acronym in individual sections, such as for ellagic acid on page 12, line 2 (where it again says [EA]).

Point 3.6 should be "Anthocyanins."

Figure 2 is very interesting. Simply labeling something as "approved" may not be sufficient, as there is a significant difference between approval as a medication and as a supplement. It would be helpful to specify the "type of approval" to provide clarity.

In the conclusion, "anthocyanins" should be in plural form

Reviewer 2 Report

Comments and Suggestions for Authors

Review of Manuscript ID nutrients-3196792

Title: Dietary Polyphenols as Anti-Aging Agents: Targeting the Hallmarks of Aging

Authors: Ying Liu, Minglv Fang, Xiaohui Tu, Xueying Mo, Lu Zhang, Binrui Yang, Feijie Wang, Young-Bum Kim, Cheng Huang, Liang Chen, Shengjie Fan.

General comment: the manuscript reviews data regarding the effect of polyphenols in the process of aging. While the idea is sound and, due to the intense research on the field of anti-aging compounds continuous updating of information is necessary, the objective is too wide and relevant data and attractive studies are missing. Particularly in the case of flavanols, a group of flavonoids that has proved very active and beneficial in the prevention of oxidative stress, inflammation, immune function and interaction with gut microbiota, that are barely mentioned. Only EGCG, which is not mentioned as a flavanol in the whole text, is included in the review. This omission should be mended before the manuscript is accepted for publication. Some specific comments are detailed below:

Specific comments:

1)      Line 85; flavanols and anthocyanins are considered as subfamilies of flavonoids, not as independent polyphenol groups.

2)      Line 92; see comment above regarding flavonoid classification.

3)      Lines 99-127; curcumin, resveratrol, apigenin and EGCG should be identified and ascribed to a particular class of polyphenols at their first appearance in text. The same applies to the rest of compounds mentioned through the rest of the manuscript.

4)      Lines 99-121; the authors have selected and exposed just a few example polyphenols, particularly resveratrol, that affect metabolism, immune system and inflammation, but there are many other phenolic compounds that have been proved effective in the same processes. As an example, see flavanol effects at Goya, L. and De Pascual-Teresa, S. Effects of Polyphenol-Rich Foods on Chronic Diseases. Editorial Nutrients 2023, 15(19), 4134.

5)      Lines 135-159, same as above; the authors focus on bioavailability of curcumin and its effect and interaction with microbiota, but there are many other polyphenolic compounds whose effect on microbiota has been studied and are not commented.

6)      Figure 1; there are many acronyms that are not spelled our nor explained.

7)      Table 1, same as for figure 1; there are acronyms and animal models such as SD rats, ddY mice, HtrA2 KO mice, etc. that need to be spelled out or introduced.

8)      Table 1; references 78, 84 and 93 are not included in the table nor mentioned in the text.

9)      Table 1, effects of naringenin; reference 90 quoted deals with anti-aging effects of anthocyanins on neural stem cells and aging mice, whereas naringenin is a flavanone.

10)   Figure 2; there are many acronyms not previously introduced and compounds such as aspirin and rapamycin that are not polyphenols.

11)   Lines 145-146; sentence need to be rewritten, since histones are not produced to counteract or quench ROS.

12)   Line 228; the text states the effect of quercetin on AD animal models whereas reference 148 quoted at the end deals with cultured PC-12 cells.

13)   Line 288; reference 161 is the same as number 4.

14)   Line 432; the text mentions effects of ellagic acid while the reference quoted (194) deals with grape seed extract.

15)   References 65 and 66; titles are incomplete, and reference 161 is repeated, same as reference 4.

Comments on the Quality of English Language

See comments above

Reviewer 3 Report

Comments and Suggestions for Authors

The work is interesting, well-written and worth publishing. Tables well constructed, interesting ending.

My comments:

1/ a list of abbreviations is necessary

2/ please add more graphics and ensure their high quality to make the manuscript more attractive

3/ please revise and standardize the spelling of phenolic compounds

Comments on the Quality of English Language

In my opinion, professional language proofreading needed to make manuscript easier to read

Round 2

Reviewer 2 Report

Comments and Suggestions for Authors

The authors have conveniently addressed all my comments and queries; thus, I suggest to accept the revised version 1 for publication at Nutrients.